# Pathophysiological Perspective of Osteoarthritis

**DOI:** 10.3390/medicina56110614

**Published:** 2020-11-16

**Authors:** Mohd Heikal Mohd Yunus, Abid Nordin, Haziq Kamal

**Affiliations:** 1Department of Physiology, Universiti Kebangsaan Malaysia Medical Centre, Kuala Lumpur 56000, Malaysia; m.abid.nordin@gmail.com (A.N.); kamalntee1@gmail.com (H.K.); 2Centre of Tissue Engineering & Regenerative Medicine, Universiti Kebangsaan Malaysia Medical Centre, Kuala Lumpur 56000, Malaysia

**Keywords:** osteoarthritis, pathogenesis, cytokines, proteolytic enzymes

## Abstract

Osteoarthritis (OA) is the most well-known degenerative disease among the geriatric and is a main cause of significant disability in daily living. It has a multifactorial etiology and is characterized by pathological changes in the knee joint structure including cartilage erosion, synovial inflammation, and subchondral sclerosis with osteophyte formation. To date, no efficient treatment is capable of altering the pathological progression of OA, and current therapy is broadly divided into pharmacological and nonpharmacological measures prior to surgical intervention. In this review, the significant risk factors and mediators, such as cytokines, proteolytic enzymes, and nitric oxide, that trigger the loss of the normal homeostasis and structural changes in the articular cartilage during the progression of OA are described. As the understanding of the mechanisms underlying OA improves, treatments are being developed that target specific mediators thought to promote the cartilage destruction that results from imbalanced catabolic and anabolic activity in the joint.

## 1. Introduction

Osteoarthritis (OA) is the most common cause of chronic joint pain among the geriatric population. OA is defined as the progressive deterioration of articular cartilage, followed by inflammation in the synovial cavity. Due to the extreme pain in the joint caused by OA, patients experience significant disability in their daily living. 

Prior to the 1990s, OA had been described as cartilage wear and tear, where the articular cartilage is degraded due to incremental pressure on a particular joint. With the advancement of molecular biology, the paradigm of OA pathophysiology has shifted to it being described as an inflammatory joint disease [1,2,3]. This follows the discovery of several inflammatory mediators that actuate chondrocytes to produce matrix metalloproteinases (MMPs), a major player in articular matrix degradation.

In more recent years, the establishment of a direct correlation between age-related inflammation and the disturbance in gut microbiota has brought attention to the gut–joint axis hypothesis of OA. The link between the disturbance in gut microbiota, defined as gut dysbiosis, and OA has been demonstrated in several studies [4].

Maintenance of the articular cartilage is tightly regulated by the anabolic and catabolic pathways of the cartilage matrix. In a healthy joint, the articular chondrocytes adapt to the various stresses to which they are subjected by altering their metabolism, resulting in the degradation or synthesis of the cartilage matrix to suit the demands of the body [5,6,7].

The complex pathogenesis of OA comprises the interplay of numerous factors ranging from hereditary inclination to alteration of gene expression via changes in the mechanical loading experienced by articular chondrocytes [5]. Dysregulation in these molecular repertoires can prompt the deterioration of the articular cartilage and the risk of progression into OA, either directly or indirectly [1].

## 2. Clinical Features

The main symptoms of OA are pain, joint stiffness, joint impairment, and reduced range of motion. 

### 2.1. Pain

At clinical presentation, the earliest and most common indicator of OA progression is chronic pain in the knee joint. Although not completely understood, hypotheses on the origin of the pain include the nociceptor fibers and the mechanoreceptors in the subchondral bone and synovial cavity [8]. It has been suggested that increased concentrations of excitatory amino acids (EAA), particularly glutamate, released from sensory neurons in the spinal cord lead to the hyperalgesia and pain in the influenced region [9].

The origin of pain is also hypothesized to be due to bone friction when the cartilage is no longer able to maintain the normal distance between two bones. Termed joint space narrowing, it is indicated by the loss of radiolucent cartilage with the appearance of whitening of the subchondral bone under plain X-ray [10]. 

In addition to joint space narrowing, the precise mechanical causes of pain in OA include osteophytes growth with stretching of the periosteum, increased intra-osseous pressure, subchondral microfractures, ligament damage, capsular tension, meniscal injury, and synovitis. [11]. Table 1 shows the stages of pain in OA according to Hawker G. [12].

### 2.2. Joint Stiffness

Joint stiffness is a typical symptom in OA. Joint stiffness may be represented as difficulty or discomfort during movement because of perceived inflexibility of the joint. Deficiency of surface-active phospholipid (SAPL), the synovial surfactant, plays a prominent role in joint stiffness [13]. Stiffness is generally most observable immediately in the first part of the day, yet it may likewise occur later in the day, particularly after periods of inactivity. In patients with OA, both morning and idleness-related stiffness rapidly improve and resolve, but the joint pain gradually exacerbates with frequent use [8].

### 2.3. Bone Enlargement and Swelling

OA results in enlargement and swelling of the bone, which may sometimes be visible in both smaller joints such as the interphalangeal joints and larger joints such as the knee. Bone swelling occurs due to numerous pathological changes that take place during OA. Among the changes are soft tissue oedema, blockage of blood circulation, damaged chondrocytes, increased bone density, and the formation of cystic changes [14]. Together, these pathological changes trigger bone remodeling, leading to a variety of outcomes such as marginal osteophytosis, joint subluxation, capsular thickening, synovial hyperplasia and synovial effusion. 

In combination, these changes to the bone structure contribute to the reduced range of both active and passive movements in patients [14]. In severe cases, the lack of movement can lead to fixed flexion deformity at large joints such as the knees, hips, or elbows [10].

## 3. Risk Factors of Knee OA

OA has a multifactorial etiology. It may be thought of as an endpoint outcome of a crosstalk between local and systemic factors. 

### 3.1. Aging

Considering that OA is most common among the elderly, increasing age is named as the most prominent risk factor for its development [15]. Aging drives changes in the joint tissues, making the joint increasingly susceptible to the development and progression of OA over time. Modification of the mechanical properties of the cartilage, influenced by rearrangement of the extracellular matrix (ECM), accumulation of advanced glycation end-products (AGEs), decreased aggrecan size, diminished hydration, and expanded collagen cleavage, lead to its increased susceptibility to degeneration [16]. Meanwhile, in chondrocytes, mitochondrial abnormalities, oxidative stress, and diminished autophagy alter their capacity, stimulating the catabolic pathway and cell death [17].

### 3.2. Joint Injury and Trauma

Articular cartilage is a durable tissue, capable of enduring the repetitive stress produced from the daily physical activities. However, it remains susceptible to trauma that can damage the cartilage and subchondral bone. Such damage, along with intra-articular fracture, can increase the risk of OA progression [18]. The pathologic changes are frequently evident within 10 years after injury, with the time of beginning affected to some extent by the patient’s age at the time of injury [19]. The presence of elevated host inflammatory mediators, including interleukin-6 (IL-6) and tumor necrosis factor alpha (TNF-α), and the degradation of collagen and proteoglycan after injuries involving the joint initiate the OA process [20].

### 3.3. Obesity

Obesity has a direct and indirect effect on OA. Increased body weight, indicated by elevated body mass index (BMI) in obese patients, results in significant overloading and injury to the weight-bearing joint [21]. 

Additionally, elevated BMI also results in metabolic abnormalities indicated by the leptin and adiponectin production by adipocytes within adipose tissue that have been associated with direct effects on the joint tissues that promote the development of OA. The proinflammatory cytokines produced by macrophages, i.e., IL-6 and TNF-α, have been implicated in the promotion of the proinflammatory state during OA [22]. 

The two paradigms of the relationship between elevated BMI and OA were apparent in the Netherlands Epidemiology of Obesity study, whereby knee OA was associated with weight and fat-free mass, adjusting for metabolic factors, and hand OA was associated with the metabolic syndrome, adjusting for weight [23].

### 3.4. Genetics

Epidemiological studies with twins revealed that 39–65% of OA cases in the general population can be attributed to genetic factors [24]. Hereditary forms of OA because of certain uncommon mutations in type II, IX, or XI collagen, common collagens found in articular cartilage, result in premature OA that can begin as early as adolescence, bringing about a severe, destructive form of arthritis that influences various joints [25]. However, the evidence connecting genetic factors with OA of the lower extremity joints such as the knee or hip is less conclusive in comparison to that for OA of the hands [26]. 

### 3.5. Anatomic Factors

The shape of the joint can influence the development of OA. A significant anatomic factor identified with knee OA is lower extremity alignment. Moreover, other factors that can increase the risk for OA development and progression in the knee include a leg length discrepancy of ≥1 cm, varus and valgus deformities, and tearing of the cruciate ligament [10]. Individuals with either varus alignment (bow-legged) or valgus alignment are at increased risk of tibiofemoral OA [27]. 

The relationship of anatomic factors to OA is best clarified by altered joint mechanics as the initiating cause for OA. Altered mechanics that place extreme and abnormal burdens on joint tissue cells initiate the mechanotransduction pathways that result in increased secretion of inflammatory mediators and proteolytic enzymes [28]. 

The recent increase in ankle OA diagnosis revealed the rarity of anatomic factors-induced ankle OA incidence. The occurrence of ankle OA is almost exclusively due to a preexisting fracture [29]. This observation could be attributed to the scarcity of reports on ankle OA.

### 3.6. Demographics

Females have a higher risk of developing OA. The incidence rate of OA in women aged ≥65 years is 68% as compared to 58% among men aged ≥65 years. The strong association of OA with age could explain why OA is more common in the postmenopausal years. Postmenopausal women are more susceptible to knee arthritis because of their increased levels of calcitonin and bone resorption. However, there is some evidence that the loss of estrogen could be a contributing factor [30]. 

Association of ethnicity and OA is well-established. In the United States, Caucasians demonstrated lower prevalence of OA in comparison to other ethnicities, such as African Americans, Chinese and Hispanics [31]. These differences may be contributed to by the differences in bone resorption between different ethnic groups, which are consistently reported in epidemiological studies of bone health [32]. This is supported by the identification of specific radiographic differences in some features of osteoarthritis according to ethnicity [33]. Alternatively, cultural practice and socioeconomic factors subjected to a certain ethnic group have also been postulated to affect the difference in OA prevalence [34]. 

### 3.7. Gut-Joint Axis

The association of gut dysbiosis and OA was established when quantitative and qualitative alterations to the gut microbiota (GM) demonstrated a sustained, low-grade, and chronic systemic inflammation, subsequently manifested in OA [35]. In an undisturbed state, the GM performs several functions such as nutrient absorption, maintenance of metabolic homeostasis, protection from infections, and development of systemic and mucosal immunity. In gut dysbiosis, perturbation of the GM resulted in perturbation of immune response and the host metabolism. Together, these disruptions exacerbated OA pathophysiology.

## 4. Pathological Changes in OA

Considering its complexity, the initiation, progression, and severity of OA are each driven by a plethora of factors. Furthermore, in all individuals, OA does not progress at a similar rate. At the cartilage–bone interface, an inverse relationship between subchondral bone changes and articular cartilage degeneration has been reported. As the subchondral bone thickens, a higher stage of cartilage degeneration is observed [36]. 

The earliest pathological changes in OA are commonly seen on the articular cartilage surface, with fibrillation occurring in focal regions experiencing maximal load. The proliferation of chondrocytes, the only cell type present in cartilage, dramatically accelerates in response to the loss of matrix. Some chondrocytes undergo a phenotypic change to hypertrophic chondrocytes, which is similar to the cells found in the growth plate’s hypertrophic zones. As OA progresses, extensive matrix degradation and loss occurs due to the continuous production of proteases driven by proinflammatory cytokines, which stimulate chondrocytes to produce more cytokines and proteases in an autocrine and paracrine manner. As significant matrix damage occurs, areas of the matrix devoid of cells can be seen as a result of chondrocyte apoptosis.

The bone changes in OA include subchondral sclerosis due to increased collagen production, with osteophyte formation and bone cysts at more advanced stages. Osteophytes have been described as bone and cartilage outgrowths occurring at the joint area. The direction of osteophyte growth is sensitive to the size and local cartilage narrowing, except for the lateral tibia and medial patella [37]. Biomechanical factors support osteophyte development. Most patients with symptomatic OA exhibit synovial inflammation and hypertrophy [38]. However, synovitis inflammation is not the triggering factor for primary OA, but contributes to the progression of pain and disease [39]. 

Plain radiographs underestimate the joint tissue involvement in OA, since they only visualize a component of the condition including cartilage loss that result in joint space narrowing and bony changes that result in subchondral sclerosis, cysts, and osteophyte formation. Once these changes are apparent on radiographs, the condition has significantly advanced [40]. 

Magnetic resonance imaging (MRI) studies can detect early disease and have provided evidence of matrix changes in cartilage, synovitis, bone marrow lesions, and degenerative changes in soft-tissue structures beyond the cartilage including ligaments and the knee menisci [41].

The arthroscope can play an important diagnostic role in patients with unexplained knee pain and swelling or in patients with established knee arthritis whose symptoms are disproportionate to radiographic findings [42].

Moreover, apart from these above mentioned pathological changes, the paradigm has shifted to the involvement of various inflammatory mediators, proteinases, cell proliferation, and biochemical parameters in the development of the disease.

## 5. Inflammatory Mediators

### 5.1. Cytokines and Chemokines

Inflammatory mediators such as cytokines are the key component of most inflammatory processes. Accordingly, a multitude of cytokines have been associated with OA pathogenesis. In OA patients, cartilage matrix homeostasis is disrupted by proinflammatory cytokines and chemokines [43,44]. Investigation of the cytokines and chemokines involved during OA progression revealed the upregulation of IL-1, IL-6, and IL-8 [45,46,47]. 

These cytokines act as both autocrine and paracrine agents, to stimulate the collective production of proteases, nitric oxide (NO), and eicosanoids such as prostaglandins and leukotrienes by macrophages and chondrocytes. Subsequently, the action of these inflammatory mediators in the cartilage results in the induction of the catabolic pathways, inhibition of matrix synthesis, and promotion of cellular apoptosis [47]. The cellular apoptosis, particularly in chondrocytes, is driven by the inhibition of autophagy by the proinflammatory cytokines [48,49]. 

The production of IL-1 by the stimulated chondrocytes in turn induces the synthesis of MMPs, namely MMP-1, MMP-3 and MMP-13. This is accompanied by the amplification of proinflammatory cytokines such as TNF-α, IL-6 and the chemokine IL-8, which magnifies the cartilage matrix breakdown effects in the catabolic cascade, further enhancing articular chondrocyte destruction [45,50]. IL-1 has also been proposed to contribute to the decline in cartilage matrix by inhibiting the synthesis of key components of ECM, such as proteoglycans, aggrecan, and type II collagen [51,52,53]. 

Moreover, the involvement of fibronectin in cartilage degradation is also apparent when fragments of the protein induce the expression of inflammatory cytokines, chemokines, and MMPs in chondrocytes [54,55]. In normal adult cartilage, chondrocytes synthesize matrix components very slowly. Finally, chondrocyte senescence is the other major contributor to OA development and progression. This is due to the senescent cells’ loss of the capacity for maintaining and repairing the cartilage ECM [56]. Both IL-6 and IL-8 are a key cytokine and chemokine, respectively, also known to be secreted by senescent cells, which is known as the senescence-associated secretory phenotype [57].

### 5.2. Proteases

The MMP family plays a major role in articular cartilage homeostasis. Collagenases (MMP-1, MMP-13) are responsible for degradation of the collagenous framework, whereas stromelysin (MMP-3) and aggrecanase (ADAMT-4), which is responsible for proteoglycan degradation, play prominent roles in ECM degradation [44,58]. The inflammatory cytokines synthesized by OA chondrocytes, i.e., IL-1 and TNF-α, can trigger increased MMP expression, suppress MMP enzyme inhibitors, and decrease ECM synthesis. Active stromelysin serves as an activator of collagenase 1, 2, and 3 (MMP-1, MMP-8, and MMP-13, respectively) implicated in type II collagen degradation [51,52,53]. 

MMP-13, the protease that preferentially degrades type II collagen, may be the most important in OA progression, considering type II collagen as the main collagen type in ECM. Indeed, MMP-13 expression is of greatly increased in OA [47,59]. In contrast to MMP-1 and MMP-3, which are present in high levels in OA synovial fluid, MMP-13 is highly expressed in OA cartilage, indicating its important role in the degradation of human articular cartilage throughout OA [58,59,60]. Moreover, only hypertrophic chondrocytes express the MMP-13 encoding genes, which can all be detected in OA cartilage [55]. 

Taken together, cytokine regulation of the equilibrium between the anabolic and catabolic processes determines the integrity of articular joint tissue. In pathogenesis, the occurrence of anabolic activity overwhelms that of catabolic activity, resulting in tissue degeneration [61]. 

### 5.3. Inflammatory Mediator Enzymes 

Other than cytokines and proteases, the expression of enzymes such as inducible NO synthase (iNOS), which generates the free radical NO, and cyclooxygenase-2 (COX- 2), which produces prostaglandin E2 (PGE2), are also altered in OA [50]. Here, the proinflammatory cytokine IL-1 stimulates the upregulation of both PGE2 and NO by inducing the gene expression or activity of COX-2 and iNOS [51]. 

Akhtar et al. (2011) noted that IL-1, together with mechanical loading of the cartilage, induced upregulation of the iNOS gene, which in turn increased the NO production. NO contributes to articular degradation by upregulating synthesis of MMP via cyclic GMP (cGMP)-dependent pathways while simultaneously inhibiting the synthesis of both proteoglycans and collagen [62,63]. 

Notably, NO has also been implicated to play a role in mediating chondrocyte apoptosis, a common feature in progressive OA [47,61,64]. Moreover, NO also alters mitochondrial function in OA chondrocytes, resulting in reduced cell survival by inhibiting the activity of the mitochondrial respiratory chain and ATP synthesis [65]. 

COX activation enhances the production of MMP-3 while inhibiting proteoglycans and collagen synthesis and inducing chondrocyte apoptosis [65]. With IL-1 stimulation, the chondrocyte is upregulated, eventually leading to increased production of PGE2 [66]. Martel-Pelletier et al. (2003) suggested the role of PGE2 in inflammation, apoptosis, angiogenesis, and probably the structural changes characterizing arthritic diseases [67]. Increased PGE2 production causes cartilage resorption by suppressing proteoglycans production, enhancing the degradation of both aggrecan and type II collagen and potentiating the effects of other inflammatory mediators such as IL-6 and MMP-13 [68]. 

Even without cytokine stimulation, when cultured in vitro, both NO and COX-2 levels are already highly expressed in chondrocytes from OA tissues [69,70]. Such metabolic changes may indicate a permanent phenotypical shift in OA chondrocytes. Moreover, the discovery of COX-2-induced PGE2 in fibrocartilage implies a role for PGE2 in the secondary remodeling of the tissue that causes osteophyte formation of in the pathogenesis of OA [71]. 

Matriptase, a novel protease found in OA articular chondrocytes, initiates cartilage matrix degradation by activating proteinase-activated receptor 2 (PAR-2) [72]. The absence of PAR-2 results in the absence of OA-associated pain and osteophyte formation [73]. Therefore, it is possible that the PAR-2 system is involved in the inflammatory response-mediated ECM degradation in OA. In addition, secreted proinflammatory cytokines up-regulate the expression of PAR-2, inducing greater production of proinflammatory cytokines (IL-6, IL-8), metalloproteinases, and PGE2 to enhance the inflammatory responses [74,75]. Boileau et al. (2007) demonstrated that PAR-2 expression and protein levels in OA chondrocytes have increased significantly and that the levels are regulated by the proinflammatory cytokine IL-1. The PAR-2 activation resulting in increased rates of MMPs (MMP-1, MMP-13) and COX-2 indicates that it could play a key role in the catabolic and inflammatory pathways during the progression of OA by inducing major catabolic and inflammatory mediators [75,76,77,78]. The interaction of various inflammatory mediators in the event of OA is summarized in Table 2 and Figure 1.

### 5.4. Other Potential Mediators of OA

Several studies have provided evidence for a number of potential mediators that induce OA but are not considered as inflammatory mediators. These factors also induce activating pathways that promote joint tissue destruction or inhibiting the ability of cells to repair damaged matrix of OA.

Hypoxia-inducible factor 1-alpha (HIF-1α) is an important mediator of cellular response towards an oxygen-deprived environment. Articular cartilage resides in an environment that is devoid of oxygen. Homeostasis of this tissue is mainly maintained by the HIF-1α regulatory mechanism. Single nucleotide polymorphism (SNP) studies have revealed that a defect in HIF-1α disrupts the catabolic pathways of the cartilage matrix. Instead of undergoing autophagy, defects of HIF-1α resulted in chondrocyte hypertrophy in response to the hypoxia environment [79]. 

The Wnt signaling pathways play a substantial role in the joint development. In OA, the interaction between the underlying subchondral bone and articular cartilage brought about the hypothesis of Wnt signaling pathways role in OA [80]. In a mouse model, activation of the Wnt signaling pathway in subchondral bone induces degradation of the articular cartilage. This reiterates the potential role of the Wnt signaling pathway in the pathogenesis of OA [81].

Nerve growth factor (NGF) is a neurotrophin that transmits the pain information following inflammation. In bovine chondrocytes induced with TGF-β1 and IL-1β, NGF expression was found to be elevated. The elevated expression of NGF is mediated by activin receptor-like kinase 5 (ALK5) and the Smad 2/3 complex [82]. When cartilage explant was incubated with osteoarthritic synovium, TGF-β1 and Smad 2/3 were inhibited, suggesting a potential inhibition of NGF [83]. In conclusion, NGF might be an important mediator to the OA event. 

## 6. Conclusions

Conservative medical treatment of OA is focused primarily on the pathophysiological events that modify the initiation and progression of OA. As the understanding of the mechanisms underlying OA improves, treatments are being developed that target specific mediators thought to promote the tissue destruction that results from imbalanced catabolic and anabolic activity in the joint. 

## Figures and Tables

**Figure 1 medicina-56-00614-f001:**
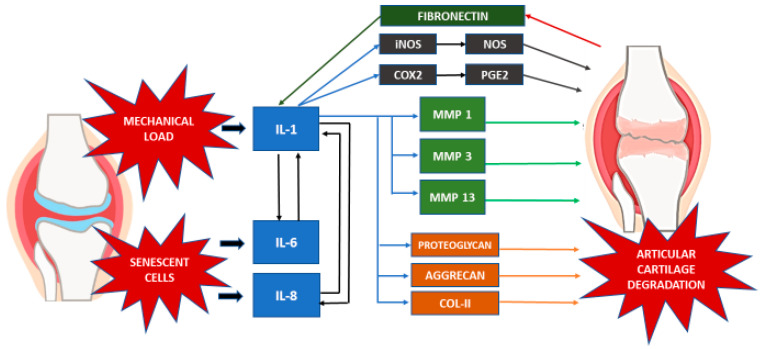
Inflammatory mediators in OA.

**Table 1 medicina-56-00614-t001:** Stages of pain in osteoarthritis (OA).

Early stage Sharp, predictable pain, normally brought about by a mechanical injury that after sometimes limits high-impact activities. Effect on function may be insignificant.
Middle stage (mild-moderate) More frequent pain with unpredictable episodes of stiffness. The pain starts to impede daily lives activity.
Advanced stages Constant throbbing pain, interspersed by short episodes of mostly unpredictable, intense, excruciating pain that severely hinder functions.

**Table 2 medicina-56-00614-t002:** Potential mechanisms in the event of OA.

Inflammatory Mediators	Description of Mechanism
Cytokines and Chemokines	IL-1, IL-6, IL-8: autocrine/paracrine agent; induce chondrocytes to produce proteases, nitric oxide, and eicosanoids such as prostaglandins and leukotrienes [34]. inhibit matrix synthesis, and promote cellular apoptosis [34,35,36]. IL-1: induce the synthesis of matrix metalloproteinases (MMP 1, MMP 3, MMP 13) TNFα, IL-6 and IL-8 to drives the cartilage matrix breakdown [32,37]. decreases the synthesis of, such as proteoglycans, aggrecan, and type II collagen [38]. IL-6 and IL-8: secreted by senescent cells, responsible for the loss of the cartilage extracellular matrix (ECM) the capability to maintain and repair [41,42].
Proteases	MMP-1, -3, -13 and *ADAMT-4:* degradation of collagenous framework and extracellular matrix [31,43]. MMP-3: activator of other collagenases (MMPs 1, 8, and 13) that implicated in type II collagen degradation [38]. MMP-13: most an important role during OA pathogenesis [34,44]. secreted by hypertrophic chondrocytes in OA cartilage [40]. degrades type II collagen as the main articular ECM [43,44,45].
iNOS (NO)	induce inhibition the synthesis of both proteoglycans and collagen [47,48]. upregulate the synthesis of matrix metalloproteinases [47,48]. induce chondrocyte apoptosis; reduced the survival of cells and inhibited mitochondrial respiratory chain function and ATP synthesis [34,38,46,49].
COX-2 (PGE2)	suppress the production of proteoglycans, enhances the degradations of both aggrecan and type II collagen [50,53]. involves inflammation, apoptosis, angiogenesis [52]. enhances the effects of IL-6, MMP-3 and MMP-13 [50,53].
PAR-2	induces pain and osteophytes formation [58]. induces production of IL6, IL8, MMPs (MMP1, MMP13) and PGE2 to enhance inflammatory responses [59,60,61,62,63]

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
