# Peer review of "Pathophysiological Perspective of Osteoarthritis"

_medicina, 2020, doi:10.3390/medicina56110614_

Round 1
Reviewer 1 Report
Dear Authors,
in this review, you aim to describe the potential risk factors and mediators implicated in the development of OA.
The topic is very interesting and the article is well written.
On the other hand, I suggest some minor revisions to improve this work.
Minor revisions
"Introduction and Risk Factors Sections": The emerging concept of the "gut-joint axis" should be considered, describing the role of gut microbiota in development of OA. Thus, please improve these sections, clarifying that a leaky gut syndrome could affect also joint tissue. Additional references recommended:
- de Sire, A.; de Sire, R.; Petito, V.; Masi, L.; Cisari, C.; Gasbarrini, A.; Scaldaferri, F.; Invernizzi, M. Gut–Joint Axis: The Role of Physical Exercise on Gut Microbiota Modulation in Older People with Osteoarthritis. Nutrients 2020, 12, 574.
- , et al Cartilage-gut-microbiome axis: a new paradigm for novel therapeutic opportunities in osteoarthritis
Reviewer 2 Report
Abstract: OK
Intro: OK
Clinical features:
- 2.1: precise mechanical pain
Risk factors:
- 3.3: Obesity is also a risk factor for hand OA
- 3.4: please add a sentence on epidemiological studies in twins
- 3.5: please precise that ankle OA is very rare without preexisting fractures
- 3.6: there are strong differences in bone resorption among various ethnic
Pathological changes
- please add a few sentences on arthroscopy and MRI respective contributions.
Mediators
- the putative paracrine role of synovial tissue and synovial fluid is not mentioned, as hypoxia.
- PPAR? Wnt?, NGF? (as putative drugs or biotherapies targets)
